# Rendering with a Gut Feeling: Depth-Guided Triangle Splatting for Physically Consistent Colonoscopic Reconstruction

**Romain Hardy**[1] 🆔                                    ROMAIN_HARDY@FAS.HARVARD.EDU

**Andrea Dunn Beltran**[2]                                    ASDUNNBE@CS.UNC.EDU

**Todd A. Brenner**[3]

**Tyler M. Berzin**[3]

**Pranav Rajpurkar**[1]

[1] *Harvard University, Department of Biomedical Informatics, Boston, MA, USA*

[2] *The University of North Carolina, Department of Computer Science, Chapel Hill, NC, USA*

[3] *Beth Israel Deaconess Medical Center, Boston, MA, USA*

**Editors:** Accepted for publication at MIDL 2026

## Abstract

Colonoscopy scene reconstruction under monocular imaging remains challenging due to affine depth ambiguity in geometric priors and strong viewpoint-dependent specularities from coaxial illumination. We present **GutSee**, a depth-guided triangle splatting framework that addresses these challenges through two key innovations. First, we introduce an *affine-invariant* depth supervision scheme that accounts for per-frame scale and shift ambiguities in pretrained monocular depth estimators, enabling them to provide stable geometric guidance even when their predictions are mutually inconsistent. Second, we incorporate a physically motivated illumination model with an explicit coaxial spotlight and learnable BRDF parameters, preventing specular highlights from being misinterpreted as geometry. Together with triangle primitives that naturally enforce surface continuity, these components yield reconstructions that are both geometrically faithful and photometrically realistic. On a phantom colonoscopy dataset, GutSee reduces mean depth RMSE by 16.1% over the next-best method under biased supervision while maintaining comparable rendering quality. These results demonstrate that coupling affine-invariant depth guidance with physically accurate lighting models improves resilience to supervision bias, enabling reliable reconstruction even when using imperfect depth priors.

**Keywords:** Endoscopy, 3D reconstruction, neural rendering.

## 1. Introduction

Colonoscopy is the clinical gold standard for colorectal cancer screening, yet endoscopists must infer complex anatomy from monocular video alone (Rex et al., 2017). In the absence of explicit three-dimensional (3D) cues, perceptual judgments remain qualitative and error-prone, contributing to incomplete mucosal coverage, imprecise lesion localization, and unreliable size estimation (Van Rijn et al., 2006; O'Connor et al., 2016; Schoen et al., 1997; Anderson et al., 2016). Robust 3D reconstruction offers a means to mitigate these limitations by providing navigational context (Wang et al., 2024b), supporting quantitative assessment (Abdelrahim et al., 2022; Soriero et al., 2022), and enabling comprehensive surface mapping (Ma et al., 2019; Thompson et al., 2016). However, the colonic environment differs substantially from conventional computer vision settings (Zhou et al., 2024).

Monocular reconstruction in colonoscopy is challenging for several reasons. Scale is inherently ambiguous without additional constraints (Wang et al., 2023). Tissue deformation arising from insufflation and peristalsis disrupts geometric consistency, and mucosal textures are often repetitive or locally feature-poor, complicating correspondence-based approaches (Ma et al., 2021; Schmidt et al., 2024). Long exploratory trajectories further exacerbate drift, yielding accumulated inconsistencies over the course of an examination (Pyykölä et al., 2024; Yao et al., 2021; Golhar et al., 2025). Illumination adds yet another layer of difficulty: the coaxial light source generates strong, pose-dependent specular reflections that violate the Lambertian assumptions underlying most photometric objectives (Shandro et al., 2020).

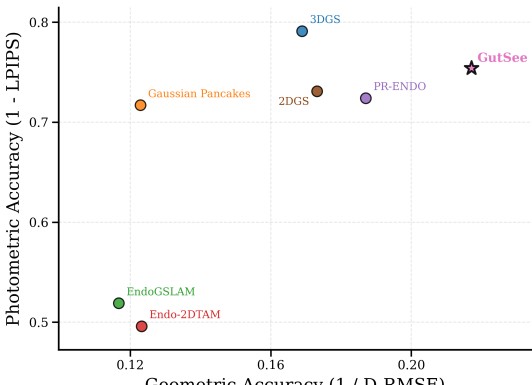

Figure 1: GutSee substantially improves geometric reconstruction accuracy under biased priors while preserving strong photometric fidelity.

Differentiable scene representations such as Neural Radiance Fields (Mildenhall et al., 2021) and Gaussian Splatting (Kerbl et al., 2023) have recently advanced general-purpose 3D reconstruction (Yan et al., 2024). These models recover geometry by minimizing photometric discrepancies between rendered and observed frames, but their flexibility can allow view-dependent appearance and density fields to absorb modeling errors, yielding photorealistic renderings with inaccurate underlying geometry (Huang et al., 2024; Wang et al., 2024a). In clinical endoscopy, where downstream utility depends on accurate surface reconstruction rather than appearance alone, such behavior is problematic.

A common approach to improving geometric fidelity is to incorporate depth supervision, typically from dedicated sensors or annotated datasets (Wang et al., 2025). In endoscopy, however, acquiring metric depth in vivo is infeasible, and even phantom datasets require careful calibration. Monocular depth networks pretrained on large-scale data offer an attractive alternative (Yang et al., 2024; Hardy et al., 2025), as they provide geometrically coherent *relative* depth. Their predictions, though, are only defined up to an affine transformation. Directly supervising rendered depth with such signals can therefore introduce undesirable bias or instability, motivating the need for a principled mechanism to exploit affine-ambiguous priors without imposing incorrect scale or shift.

A second obstacle is the entanglement of lighting and geometry in photometric losses. Standard diffuse reflectance models cannot explain the prominent, viewpoint-dependent specularities induced by coaxial endoscope lighting (Psychogyios et al., 2023; Batlle et al., 2023; Han and Wu, 2025). As a result, reconstruction methods may inadvertently encode specular highlights as geometric features unless illumination is modeled explicitly.

We introduce **GutSee**, a depth-guided triangle splatting framework designed to overcome these challenges in colonoscopic reconstruction. By combining differentiable triangle primitives with a physically based illumination model, GutSee recovers coherent surface ge-

ometry and realistic appearance from monocular video and affine-ambiguous depth priors. Our contributions are:

- An affine-invariant supervision scheme for rendered depth and normals that enables effective geometric guidance from pretrained monocular depth estimators, tolerating residual frame-wise scale and shift inconsistencies in the prior.

- A physically motivated lighting model based on a bidirectional reflectance distribution function (BRDF) with learnable material parameters and an explicit coaxial spotlight, which accounts for endoscope-specific illumination and prevents specular reflections from introducing spurious geometric detail.

- State-of-the-art geometric reconstruction accuracy on a phantom colonoscopy benchmark among methods supervised with biased depth priors, reducing the gap to methods supervised with ground-truth metric depth while maintaining high-fidelity novel view synthesis.

## 2. Related Work

**Explicit Differentiable Representations.** Recent neural rendering methods increasingly rely on explicit primitives that can be rasterized efficiently while remaining differentiable. 3D Gaussian Splatting (3DGS) (Kerbl et al., 2023) exemplifies this trend, representing scenes as anisotropic Gaussians jointly optimized for appearance and geometry. While effective for view synthesis, the infinite support of Gaussians can blur fine structures and introduce view-inconsistent surfaces, motivating the use of primitives with tighter spatial support—such as 2D Gaussians (Huang et al., 2024), smooth convex shapes (Held et al., 2025b), and Triangle Splatting (Held et al., 2025a), which offers mesh-compatible geometry with well-defined normals. We build on Triangle Splatting, whose explicit surface parameterization supports geometric priors and physically consistent illumination modeling.

**Depth Supervision Under Ambiguous Geometry.** Monocular endoscopy poses fundamental difficulties for geometric recovery, as metric depth is generally unavailable. Existing neural rendering approaches address this by supervising geometry with learned priors (Bonilla et al., 2024; Ma et al., 2021), but the resulting reconstructions inherit the biases of these signals. Real-time systems such as EndoGSLAM (Wang et al., 2024b) mitigate drift through SLAM-style tracking and structured map updates, yet remain vulnerable to errors originating from imperfect depth supervision. Our work introduces affine-invariant depth supervision to reconcile per-frame scale and shift ambiguities, allowing relative depth structure to guide optimization without the instability of metrically constrained but biased losses.

**Illumination Modeling for Near-Field Imaging.** A second obstacle to reliable geometric recovery is the near-field, coaxial illumination characteristic of endoscopy. Strong specularities and radial attenuation systematically violate the distant-light assumptions used in many photometric objectives, causing renderers to misinterpret lighting artifacts as geometry. Prior efforts incorporate endoscope-specific lighting (Kaleta et al., 2025; Han and Wu, 2025), but they depend on accurate geometry or MLP-based irradiance estimation and

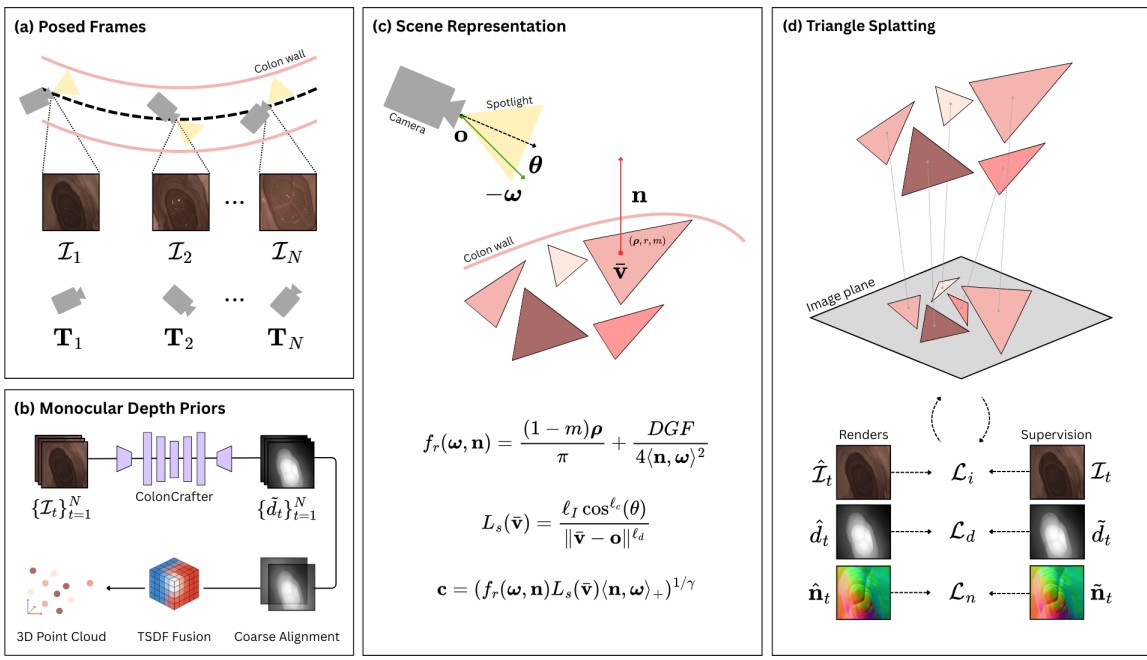

Figure 2: **Overview of GutSee.** **(a)** Given a sequence of posed RGB colonoscopy frames, **(b)** we estimate disparity maps using ColonCrafter ([Hardy et al., 2025](#)) and fuse them into an initial point cloud via TSDF integration. **(c)** The scene is represented as a set of triangle primitives with learnable material properties, rendered under a physically-based illumination model that combines a coaxial spotlight with low-frequency ambient lighting. **(d)** Triangle geometry, appearance, and lighting are optimized using photometric losses and affine-invariant geometric supervision against the biased ColonCrafter priors.

therefore remain sensitive to biased priors. We address this complementary failure mode by integrating a learnable BRDF and explicit coaxial spotlight into our splatting framework, enabling correct attribution of view-dependent effects and improving geometric robustness under imperfect supervision.

## 3. Methods

Given RGB colonoscopy frames $\{\mathcal{I}_t\}_{t=1}^N$ with corresponding camera poses $\{\mathbf{T}_t\}_{t=1}^N$, we reconstruct the colonic surface by optimizing a set of small triangles to match both the observed images and depth priors from a pretrained monocular estimator. The triangles carry material properties that interact with an explicit model of the endoscope's coaxial light source, allowing specular highlights to be explained by lighting rather than absorbed into geometry. Because monocular depth is only recoverable up to an unknown scale and shift, we supervise geometry using an affine-invariant loss that aligns rendered and predicted depths before computing residuals.

We describe the triangle-based scene representation (Section 3.1), the physically based illumination model (Section 3.2), and the optimization procedure with affine-invariant supervision (Section 3.3). An overview is shown in Figure 2.

### 3.1. Scene Representation

We represent the colon surface as a collection of small, overlapping triangles rendered via differentiable splatting. Each triangle stores its geometry (vertex positions), transparency, and material properties (color, roughness, shininess). During rendering, triangles are projected onto the image plane with soft edges that enable gradient-based optimization, then composited front-to-back to produce the final image. Because each triangle defines a well-posed local surface patch, this representation naturally provides explicit surface normals, which will be used in our physically based illumination model. Additionally, unlike Gaussian splatting methods whose kernels have infinite spatial support, triangle splats have compact, bounded support, reducing ambiguity about where the reconstructed surface lies.

Formally, we modify the Triangle Splatting framework (Held et al., 2025a) and represent the scene as a set of triangle primitives $\{T^{(k)}\}_{k=1}^{K}$. Each triangle $T^{(k)}$ is parameterized by vertices $\{\mathbf{v}_i^{(k)}\}_{i=1}^{3}$, an opacity $\alpha^{(k)}$, a smoothness parameter $\sigma^{(k)}$, and material attributes: diffuse albedo $\boldsymbol{\rho}^{(k)}$, roughness $r^{(k)}$, and metallicity $m^{(k)}$. Rendering proceeds by projecting vertices to image space via a pinhole camera. Each triangle contributes a soft footprint given by a differentiable window function $w^{(k)}(\mathbf{u}; \sigma^{(k)})$ depending on the signed distance from pixel $\mathbf{u}$ to the projected triangle, and final pixel intensities are obtained by front-to-back alpha compositing. The geometric normal

$$\mathbf{n}^{(k)} = \frac{(\mathbf{v}_2^{(k)} - \mathbf{v}_1^{(k)}) \times (\mathbf{v}_3^{(k)} - \mathbf{v}_1^{(k)})}{\|(\mathbf{v}_2^{(k)} - \mathbf{v}_1^{(k)}) \times (\mathbf{v}_3^{(k)} - \mathbf{v}_1^{(k)})\|}$$

of $T^{(k)}$ is flipped to face the camera for view consistency, and shading is evaluated at the triangle centroid $\bar{\mathbf{v}}^{(k)}$.

### 3.2. Endoscopic Illumination Model

Standard rendering models assume distant or ambient lighting, but endoscopes carry a coaxial light source that moves with the camera, producing characteristic radial falloff and view-dependent specular highlights. Following prior physically based rendering formulations, we treat the endoscopic light as a camera-mounted spotlight and describe surface reflectance using a microfacet BRDF. In particular, we adapt a Disney-style BRDF parameterization (Burley and Studios, 2012), drawing inspiration from Han and Wu (2025) and Yao et al. (2022). By modeling material reflectance and illumination separately, the optimization avoids confusing specular highlights with changes in surface geometry.

The illumination has two components. The spotlight intensity at a surface point falls off with both distance from the camera and angle from the optical axis, controlled by learnable parameters. The surface reflection combines a diffuse term (light scattered equally in all directions, controlled by albedo) and a specular term (mirror-like reflection concentrated around the reflection direction, controlled by roughness and metallicity). Because

the light source and camera are co-located, incoming and outgoing light directions coincide, simplifying the reflection computation.

Formally, given camera center $\mathbf{o}$, the viewing direction at centroid $\bar{\mathbf{v}}$ is

$$\boldsymbol{\omega} = \frac{\bar{\mathbf{v}} - \mathbf{o}}{\|\bar{\mathbf{v}} - \mathbf{o}\|}.$$

The spotlight radiance is

$$L_s(\bar{\mathbf{v}}) = \frac{\ell_I \cos^{\ell_c}(\theta)}{\|\bar{\mathbf{v}} - \mathbf{o}\|^{\ell_d}},$$

where $\theta$ is the angle to the optical axis and $\ell_I, \ell_c, \ell_d > 0$ are learnable intensity, angular falloff, and distance attenuation exponents. The surface BRDF is

$$f_r(\boldsymbol{\omega}, \mathbf{n}) = \underbrace{\frac{(1-m)\,\boldsymbol{\rho}}{\pi}}_{\text{diffuse}} + \underbrace{\frac{D(r, \mathbf{n}, \boldsymbol{\omega})\,G(r, \mathbf{n}, \boldsymbol{\omega})\,F(\boldsymbol{\omega}, \mathbf{n}, m, \boldsymbol{\rho})}{4\,\langle \mathbf{n}, \boldsymbol{\omega}\rangle^2}}_{\text{specular}},$$

where $D$ is the GGX normal-distribution function, $G$ is the Smith geometry term, and $F$ is the Schlick Fresnel approximation (Yao et al., 2022). We compute the outgoing radiance as

$$\mathbf{c} = \left(f_r(\boldsymbol{\omega}, \mathbf{n}) L_s(\bar{\mathbf{v}}) \langle \mathbf{n}, \boldsymbol{\omega}\rangle_+\right)^{1/\gamma},$$

where $\gamma$ is a learnable gamma correction mapping linear radiance to the nonlinear image domain.

### 3.3. Optimization

We initialize geometry from a point cloud produced by fusing ColonCrafter (Hardy et al., 2025) depth predictions via Truncated Sign Distance Function (TSDF) integration, a volumetric method that averages signed distance values on a voxel grid to extract a coherent surface; this provides a stable initial geometry for optimization without imposing hard constraints thereafter. We then jointly optimize triangle geometry, material properties, and lighting parameters. The loss combines photometric reconstruction with affine-invariant geometric supervision.

The core challenge is that monocular depth estimators predict depth only up to an unknown scale and shift that may vary across frames. Rather than treating these predictions as metric ground truth, we align rendered depths to the predictions before computing residuals: for each frame, we solve for the scale and shift that best match rendered depth to predicted depth, then penalize deviations from the aligned target. This allows the depth prior to guide relative geometry without imposing an incorrect scale. We apply analogous supervision to surface normals derived from the aligned depths, encouraging local surface orientation to remain consistent with the prior.

Formally, the photometric loss combines pixel-wise and structural similarity terms:

$$\mathcal{L}_i = (1 - \lambda_{\text{SSIM}})\|\hat{\mathcal{I}} - \mathcal{I}\|_1 + \lambda_{\text{SSIM}}(1 - \text{SSIM}(\hat{\mathcal{I}}, \mathcal{I})).$$

For geometric supervision, we perform alignment in disparity space, which yields a well-conditioned least-squares problem and accommodates the affine ambiguity of monocular

estimators. We align each rendered disparity map $\hat{d}(\mathbf{u})$ to the ColonCrafter prior $\tilde{d}(\mathbf{u})$ by solving a per-view least-squares fit:

$$(\beta^*, \xi^*) = \arg\min_{\beta,\xi} \sum_{\mathbf{u}\in\Omega} (\beta\,\hat{d}(\mathbf{u}) + \xi - \tilde{d}(\mathbf{u}))^2,$$

where $\Omega$ denotes the set of valid pixels. We then invert the aligned disparity to obtain metric depth: $\hat{D}(\mathbf{u}) = (\beta^*\hat{d}(\mathbf{u}) + \xi^*)^{-1}$. The depth loss penalizes residuals using a robust Huber penalty $\mathcal{H}_\delta$ together with edge-aware smoothness:

$$\mathcal{L}_d = \frac{1}{|\Omega|} \sum_{\mathbf{u}\in\Omega} \mathcal{H}_\delta(\epsilon(\mathbf{u})) + \frac{1}{2}(|\nabla_x\epsilon| + |\nabla_y\epsilon|) + \|\beta^* - 1\|_2^2 + \|\xi^*\|_2^2,$$

where $\epsilon(\mathbf{u}) = |\hat{D}(\mathbf{u}) - \tilde{D}(\mathbf{u})|$ and the penalties on $\beta^*$ and $\xi^*$ gently bias alignment toward identity. Similarly, the normal consistency loss encourages local surface orientation to remain consistent with the depth prior, helping stabilize geometry in regions where photometric cues are weak or ambiguous:

$$\mathcal{L}_n = \frac{1}{|\Omega|} \sum_{\mathbf{u}\in\Omega} (1 - \langle\hat{\mathbf{n}}(\mathbf{u}), \tilde{\mathbf{n}}(\mathbf{u})\rangle).$$

Finally, to prevent view-dependent effects from leaking into albedo, we enforce Laplacian smoothness:

$$\mathcal{L}_\rho = \frac{1}{K} \sum_{k=1}^{K} \frac{1}{|\mathcal{N}(k)|} \sum_{l\in\mathcal{N}(k)} \|\boldsymbol{\rho}^{(k)} - \boldsymbol{\rho}^{(l)}\|_2^2, \tag{1}$$

The total objective is: $\mathcal{L} = \mathcal{L}_i + \lambda_d\mathcal{L}_d + \lambda_n\mathcal{L}_n + \lambda_\rho\mathcal{L}_\rho$.

## 4. Experiments

We evaluate GutSee on phantom colonoscopy data and compare it against existing splatting approaches. To ensure fairness, we test baselines under multiple supervision protocols and report both geometric and image-based reconstruction metrics.

**Datasets.** We benchmark GutSee on 5 phantom colonoscopy sequences sampled from C3VDv1 (Bobrow et al., 2023) and 5 from C3VDv2 (Golhar et al., 2025). To our knowledge, no 3D reconstruction approaches in the literature have evaluated their methods on C3VDv2, which introduces more challenging scenes characterized by complex camera trajectories and the presence of debris.

**Baselines.** As general baselines, we consider 3DGS (Kerbl et al., 2023) and 2DGS (Huang et al., 2024). For domain-specific baselines, we evaluate EndoGSLAM (Wang et al., 2024b), Endo-2DTAM (Huang et al., 2025), Gaussian Pancakes (Bonilla et al., 2024), and PR-ENDO (Kaleta et al., 2025). For all comparisons, we fix camera poses to their true values to isolate mapping quality from tracking performance. To ensure a comprehensive comparison, we evaluate all models under two configurations: (i) a ground truth-supervised setting, and (ii) a depth prior-supervised setting using ColonCrafter predicted depth maps. This two-tier evaluation disentangles the contributions of supervision signals and model architecture, and highlights GutSee's ability to operate robustly even with biased priors.

Table 1: Quantitative comparison of GutSee with baseline methods on 10 sequences from C3VD (aggregated over v1 and v2). GutSee achieves the strongest geometric accuracy with biased depth supervision, while maintaining competitive rendering quality. Sup.: Supervision; GT: Ground Truth; CC: ColonCrafter.

| | Method | CD ↓ | D-RMSE ↓ | PSNR ↑ | SSIM ↑ | LPIPS ↓ |
|---|---|---|---|---|---|---|
| **GT Sup.** | 3DGS (Kerbl et al., 2023) | 0.579 | 1.645 | 32.055 | 0.848 | 0.242 |
| | 2DGS (Huang et al., 2024) | 0.402 | 1.488 | 30.733 | 0.833 | 0.274 |
| | EndoGSLAM (Wang et al., 2024b) | 0.563 | 3.107 | 20.808 | 0.750 | 0.468 |
| | Endo-2DTAM (Huang et al., 2025) | 0.617 | 3.326 | 18.867 | 0.731 | 0.481 |
| | Gaussian Pancakes (Bonilla et al., 2024) | 0.513 | 7.882 | 30.654 | 0.830 | 0.296 |
| | PR-ENDO (Kaleta et al., 2025) | 0.625 | 0.909 | 32.168 | 0.848 | 0.272 |
| | GutSee (ours) | 0.522 | 1.193 | 30.143 | 0.829 | 0.263 |
| **CC Sup.** | 3DGS (Kerbl et al., 2023) | 2.141 | 5.921 | 32.986 | 0.879 | 0.209 |
| | 2DGS (Huang et al., 2024) | 2.144 | 5.774 | 31.178 | 0.858 | 0.269 |
| | EndoGSLAM (Wang et al., 2024b) | 2.980 | 8.570 | 20.313 | 0.717 | 0.481 |
| | Endo-2DTAM (Huang et al., 2025) | 1.980 | 8.116 | 18.780 | 0.702 | 0.504 |
| | Gaussian Pancakes (Bonilla et al., 2024) | 2.181 | 8.139 | 31.128 | 0.858 | 0.283 |
| | PR-ENDO (Kaleta et al., 2025) | 2.566 | 5.346 | 32.393 | 0.866 | 0.276 |
| | GutSee (ours) | 1.520 | 4.605 | 30.695 | 0.857 | 0.246 |

■ Worst    ■ Middle    ■ Best

**Metrics.** We assess reconstruction quality using both geometry- and image-based metrics. For geometric accuracy, we compute the bidirectional Chamfer distance (CD) between reconstructed and ground truth point clouds to assess global surface alignment, and the depth root-mean-square error (D-RMSE) against reference depths to quantify metric depth accuracy. For image fidelity, we report peak signal-to-noise ratio (PSNR), structural similarity index measure (SSIM), and learned perceptual image patch similarity (LPIPS). Following standard practice, evaluations are performed on every eighth held-out frame. All metrics are computed per sequence and subsequently averaged across sequences.

**Implementation Details.** Across experiments, all frames are undistorted and resized to $384 \times 384$ pixels, with camera intrinsics scaled accordingly. We optimize each scene for 15,000 iterations. Densification and pruning follows the logic of the original Triangle Splatting implementation; we prune and densify triangles every 500 iterations, starting from iteration 500 and stopping at iteration 13,000. We set the initial learning rate of the triangle positions to $2 \times 10^{-3}$, and choose $\lambda_{\text{SSIM}} = 0.2$, $\lambda_d = 0.1$, $\lambda_n = 0.1$, and $\lambda_\rho = 0.1$ for the loss weights. All models are trained using PyTorch (Paszke et al., 2019) on a single NVIDIA A10 Tensor Core GPU.

**Results.** The results in Table 1 highlight two central findings. First, GutSee delivers consistently strong geometric reconstruction accuracy. With ground truth supervision, it achieves the lowest D-RMSE among all models except PR-ENDO, which we attribute to PR-ENDO's explicit regularization term that penalizes deviation from the initial (ground truth) scene geometry. Under biased ColonCrafter supervision, GutSee yields substantially lower geometric error than all baselines (1.520 mm CD, 4.605 mm D-RMSE); the next best performing model, PR-ENDO, has an average D-RMSE 0.741 mm (16.1%) greater.

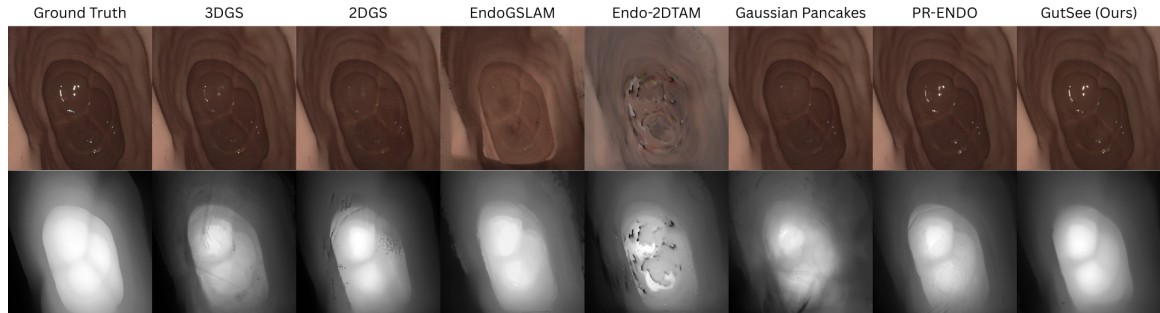

Figure 3: Qualitative comparison of novel-view renderings and depth estimates on the cecum_t1_a sequence. GutSee produces illumination-consistent renderings and depth maps with reduced artifacts and sharper geometric delineation relative to competing approaches.

Notably, GutSee maintains acceptable rendering quality with an average LPIPS value of 0.246, second only to 3DGS which overfits the images; as shown in Figure 3, the depth renders produced by 3DGS contain floating artifacts and are generally less faithful to the true geometry of the colon. Together, these results indicate that GutSee offers a robust geometry-centric reconstruction approach that does not compromise rendering quality.

Figure 3 further provides a qualitative comparison of novel-view renderings across methods on the cecum_t1_a sequence from C3VD. While Gaussian Pancakes and 3DGS recover diffuse details well, their renders exhibit muted or spatially imprecise specular highlights. PR-ENDO models illumination through an MLP, enabling more faithful specular effects, yet it can struggle with fine details and produces comparatively noisy depth maps. In contrast, GutSee produces smooth surface reconstructions while simultaneously being able to reproduce complex endoscopic lighting phenomena, even when trained under biased ColonCrafter priors.

Another advantage of our approach lies in its explicit separation of lighting and surface geometry. Because our formulation employs a physically grounded BRDF parameterization with learnable materials and an explicit light source, the method can successfully disentangle illumination effects from underlying shape. As shown in Figure 4, specular highlights emerge directly from the interaction between the estimated light source and the reconstructed surface, rather than being absorbed into material parameters or inadvertently encoded in geometry. We further examine the effect of our illumination model on reconstruction performance in Appendix D.

## 5. Discussion and Conclusion

In this work, we introduced GutSee, a depth-guided triangle splatting framework for monocular colonoscopic reconstruction that combines affine-invariant geometric supervision with an explicit model of endoscopic illumination. On phantom colonoscopy sequences, GutSee achieves lower Chamfer distance and depth RMSE than splatting-based baselines under biased supervision, while maintaining competitive photometric performance (Table 1). These

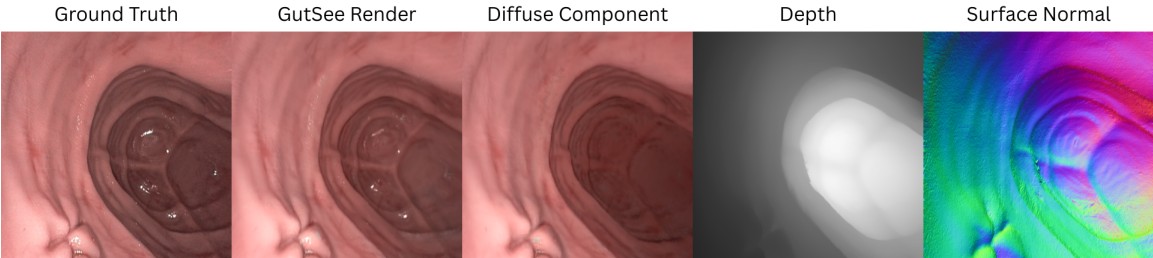

Figure 4: An example of light-geometry decoupling. Our physically based scene representation and explicit coaxial light model separates lighting effects from the underlying scene geometry.

results indicate that explicitly addressing depth ambiguity and light-geometry entanglement is an effective strategy for geometry-centric reconstruction in endoscopy. However, defining what constitutes a "realistic" colonoscopic reconstruction remains non-trivial. Global geometric metrics such as Chamfer distance or depth RMSE capture overall alignment but can hide localized errors in folds, narrow segments, or regions of strong specularities. Conversely, image-based metrics may favor visually plausible but geometrically inconsistent solutions. Although GutSee improves geometric accuracy relative to baselines, subtle deviations that could matter clinically (such as local underestimation of fold height or lesion protrusion) may not be fully captured by the current evaluation protocol. As in other generative and reconstruction settings, the ultimate quality of the reconstructions should be judged by how well they support downstream tasks, including navigation, lesion localization, and quantitative measurement.

Several limitations suggest directions for future work. Our quantitative evaluation is restricted to phantom sequences and does not cover the full spectrum of in vivo variability, including strong non-rigid motion, fluids, and patient-specific variation; we focus on phantoms because they provide reliable ground-truth geometry necessary for rigorous evaluation, viewing this as a prerequisite to clinical translation. Moreover, GutSee remains coupled to the quality of the depth prior: although the affine-invariant loss prevents incorrect local affine biases from being hard-coded into the reconstruction, it does not "repair" fundamentally inaccurate priors, and severe local errors or missing structures in the ColonCrafter predictions can still lead the optimizer to suboptimal geometry. In addition, the current formulation assumes access to known camera poses; performing full SLAM with jointly optimized poses and geometry over extended, realistic trajectories remains challenging. Extending GutSee to real clinical videos, enriching the illumination and material models, and learning transferrable features across scenes are promising directions for future work.

In summary, GutSee provides a practical framework for physically consistent, geometry-aware reconstruction in colonoscopy using only monocular RGB video and pose information. By combining affine-invariant depth guidance with explicit triangle-based geometry and a coaxial lighting model, it moves reconstruction methods closer to being not only visually convincing but also geometrically reliable enough to support computer-assisted colonoscopy.

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

## Appendix A. Visualization of Learned Albedo Parameters and Geometry

Figure 5 visualizes the diffuse albedo inferred by GutSee. The model recovers material properties that are disentangled from scene illumination, yielding intrinsic parameters that support physically consistent reconstruction and rendering.

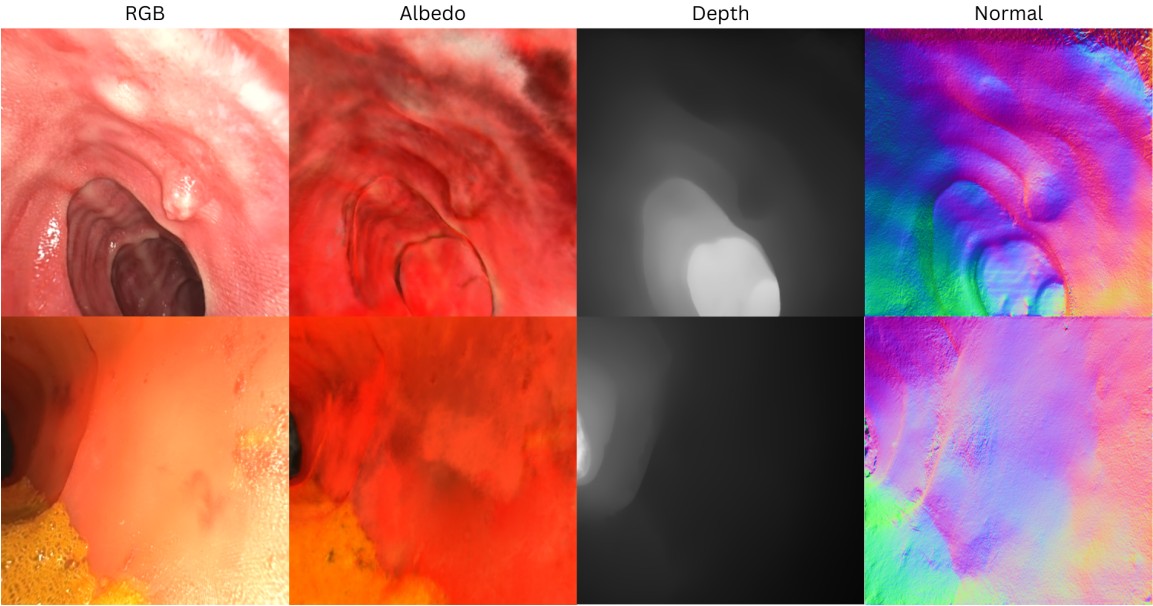

RGB      Albedo      Depth      Normal

Figure 5: Visualization of intrinsic material parameters estimated by GutSee. From left to right: held-out rendered RGB, recovered diffuse albedo, predicted depth, and estimated surface normals. The method learns illumination-invariant albedo and geometrically consistent structure from posed colonoscopy frames and biased monocular depth priors.

## Appendix B. Novel View Synthesis on Additional C3VD Sequences

Figure 6 presents supplementary qualitative comparisons of novel view synthesis on a diverse subset of C3VD scenes. Across all examples, GutSee reconstructs surfaces with high geometric fidelity while maintaining photometric quality comparable to existing baselines.

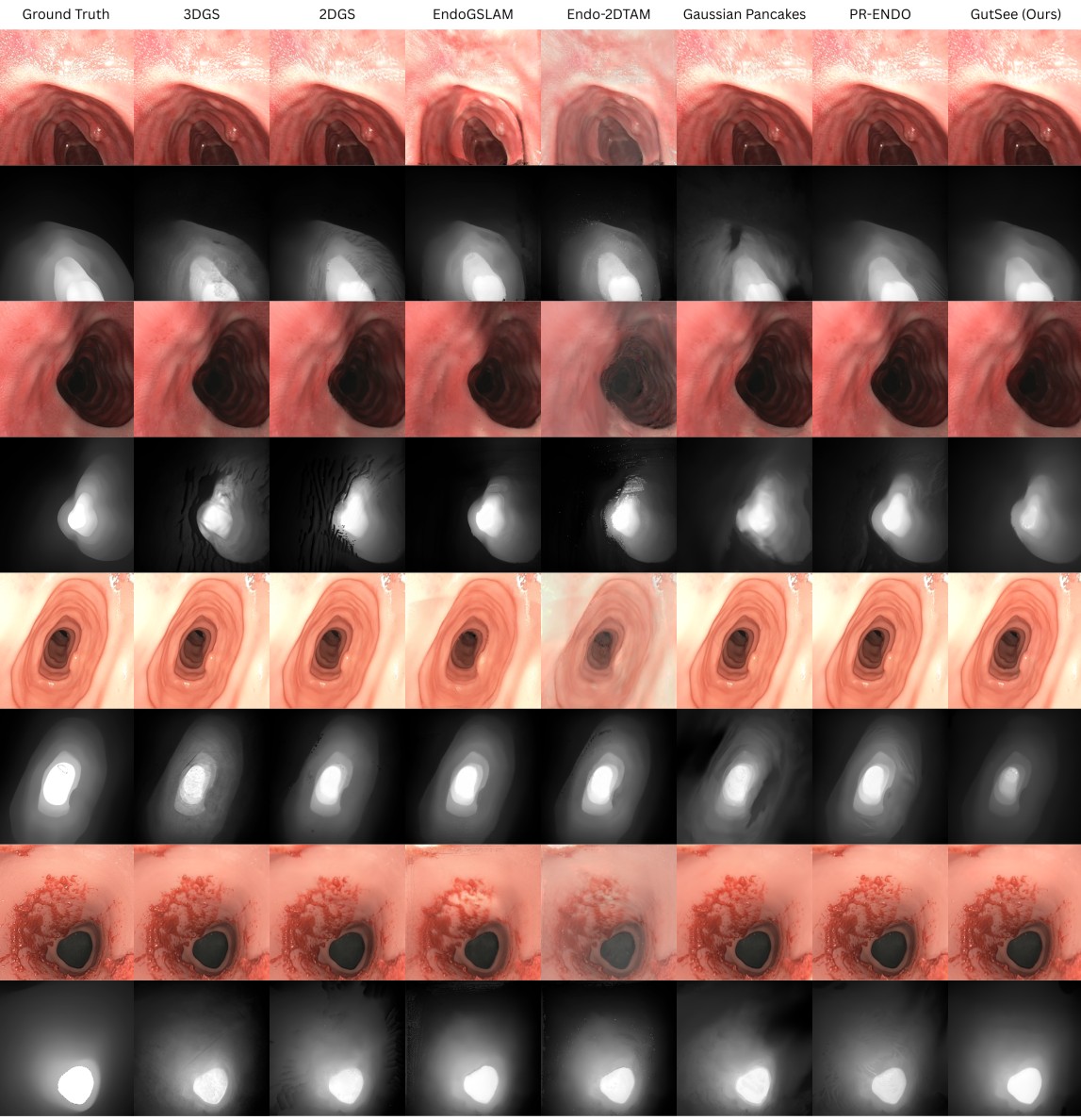

Figure 6: Additional novel-view synthesis results on representative C3VD sequences. GutSee reconstructs geometrically coherent and well-localized surfaces across varied anatomical structures, while achieving photometric accuracy at least on par with neural rendering baselines.

## Appendix C. Ablation Study: Loss Terms

We perform an ablation study on four sequences (`cecum_t1_a`, `desc_t4_a`, `c1_cecum_t4_v2`, `c2_ascending_t3_v1`) to evaluate the effect of $\mathcal{L}_d$, $\mathcal{L}_n$, and our affine-invariant formulation on reconstruction performance. Table 2 lists geometric and image reconstruction metrics for these scenes under (i) our base configuration, (ii) without explicit geometric supervision ($\lambda_d = \lambda_n = 0$), and (iii) with geometric supervision but no affine alignment step (i.e., we use the ColonCrafter priors to supervise optimization directly). Overall, our base configuration produces the most desirable results. Although photometric rendering performance improves without geometric supervision, D-RMSE increases by 3.112 mm and the model no longer renders smooth depths. Without affine alignment, CD and D-RMSE also worsen, suggesting that the model inherits bias from the ColonCrafter priors.

Table 2: Ablation study on the use of geometric loss terms and our affine alignment step. Our base configuration produces the best overall results, motivating the use of affine-invariant geometric guidance.

| | Training configuration | CD ↓ | D-RMSE ↓ | PSNR ↑ | SSIM ↑ | LPIPS ↓ |
|---|---|---|---|---|---|---|
| (i) | Base | 1.830 | 4.728 | 31.558 | 0.880 | 0.247 |
| (ii) | No geometric supervision ($\lambda_d = \lambda_n = 0$) | 2.243 | 7.840 | 32.878 | 0.892 | 0.229 |
| (iii) | No affine alignment | 2.585 | 5.480 | 31.011 | 0.875 | 0.253 |

## Appendix D. Ablation Study: Illumination Model

We perform a second ablation study on the same four sequences to evaluate the effect of our illumination model on reconstruction performance. Table 3 lists geometric and image reconstruction metrics for these scenes under (i) our base configuration, (ii) removing the spotlight model by setting $L_s = 1.0$ everywhere, (iii) removing the specular term of the BRDF function, and (iv) removing the BRDF function entirely and using the per-triangle albedos to render views. Removing the BRDF model increases D-RMSE by 0.249 mm and CD by 0.122 mm while degrading photometric accuracy (0.269 vs. 0.247 LPIPS). Removing just the specular term similarly increases D-RMSE by 0.320 mm, supporting our claim that illumination modeling improves geometric accuracy. Removing the spotlight yields slightly better geometric metrics than our base configuration (D-RMSE of 4.696 mm vs. 4.728 mm); however, this comes at a significant cost to rendering quality, with LPIPS increasing from 0.247 to 0.324. This degradation occurs because the spotlight model is necessary to accurately capture distance-dependent lighting effects.

Table 3: Ablation study on our illumination model. Removing the BRDF degrades both geometric and photometric accuracy, while the spotlight model is essential for accurate rendering.

|  | Training Configuration | CD ↓ | D-RMSE ↓ | PSNR ↑ | SSIM ↑ | LPIPS ↓ |
|---|---|---|---|---|---|---|
| (i) | Base | 1.830 | 4.728 | 31.558 | 0.880 | 0.247 |
| (ii) | No spotlight ($L_s = 1.0$) | 1.799 | 4.696 | 24.310 | 0.859 | 0.324 |
| (iii) | No specular BRDF term | 1.897 | 5.048 | 29.056 | 0.860 | 0.274 |
| (iv) | No BRDF (albedo only) | 1.952 | 4.977 | 26.768 | 0.852 | 0.269 |

## Appendix E. Ablation Study: Sensitivity to Local Depth Errors

We perform a third ablation study to evaluate the effect of local errors in the depth priors on reconstruction performance. Specifically, we introduce local perturbations in the ColonCrafter-predicted depth maps used to supervise GutSee. For a given sequence, we select frames with probability $p$; for each selected frame, we add Gaussian noise to randomly selected $32 \times 32$ patches (we choose between 1 and 8 patches per selected frame). Then, we train GutSee using the perturbed depth maps. Note that the initial point cloud and RGB images are unchanged from the base training configuration. Table 4 shows GutSee's performance under (i) our base configuration, (ii) with $p = 0.1$ (10% of depth maps are locally perturbed), (iii) with $p = 0.5$ (50% of depth maps are locally perturbed), and (iv) with $p = 0.9$ (90% of depth maps are locally perturbed). As expected, increasing the perturbation probability tends to worsen geometric accuracy. However, the performance degradation is less dependent on $p$ than we anticipated; even when $p = 0.9$, the photometric quality remains constant and the D-RMSE stays below 5 mm. This is evidence that, given a sufficiently good initialization state, GutSee is relatively insensitive to local errors in the depth maps it uses for supervision.

Table 4: Ablation study on the sensitivity of GutSee to local depth errors. Increasing the local perturbation probability decreases geometric accuracy, but GutSee remains relatively insensitive.

|       | Training configuration | CD ↓  | D-RMSE ↓ | PSNR ↑ | SSIM ↑ | LPIPS ↓ |
|-------|------------------------|-------|----------|--------|--------|---------|
| (i)   | Base                   | 1.830 | 4.728    | 31.558 | 0.880  | 0.247   |
| (ii)  | $p = 0.1$              | 1.837 | 4.751    | 31.513 | 0.879  | 0.247   |
| (iii) | $p = 0.5$              | 1.853 | 4.833    | 31.576 | 0.880  | 0.247   |
| (iv)  | $p = 0.9$              | 1.838 | 4.902    | 31.582 | 0.880  | 0.246   |

