# OpenReview forum: "Rendering with a Gut Feeling: Depth-Guided Triangle Splatting for Physically Consistent Colonoscopic Reconstruction"
_MIDL.io/2026/Conference — MIDL 2026 Poster_

### Official Review · Reviewer_pe3H · 2026-01-01

**Confidence:** 4
**Preliminary Rating:** 4
**Final Rating:** 4

**Summary:**

GutSee proposes a geometry-centric reconstruction framework for monocular colonoscopy, which is based on Triangle Splatting.

The key idea is to leverage monocular depth priors while tolerating their bias by aligning rendered and predicted disparity maps via a per-frame least-squares affine transform (Section 3.3). In addition, it adapts a bidirectional reflectance distribution function (BRDF) to model endoscope-specific illumination (i.e. specular reflections).

The method is evaluated on phantom colonoscopy datasets (C3VDv1/v2) under both ground-truth depth supervision and biased monocular depth supervision (from an off-the-shelf model). The results show that GutSee achieves the lowest depth RMSE and Chamfer distance among baselines in the biased-prior setting while maintaining competitive rendering quality.

By explicitly decoupling geometry, illumination, and ambiguous depth priors, GutSee yields more reliable surface reconstruction than prior neural rendering approaches, which is particularly significant for medical imaging scenarios where accurate geometry matters more than purely photorealistic appearance.

**Strengths:**

- The paper aims to solve a clearly defined and significant problem in monocular medical image reconstruction: how to use ambiguous depth prior to achieve an accurate geometry estimation.
- The proposed affine-invariant depth supervision, combined with an explicit triangle-based surface representation and endoscopic illumination model, is technically sound and addresses the drawback of prior neural rendering and splatting approaches that entangle lighting, appearance, and geometry.
- Experiments are thorough and convincing. Baselines include 3DGS and SOTA domain-specific neural rendering methods and two supervision settings (ground truth depth map vs model-predicted depth map). The proposed method shows robust geometry estimation (lowest CD and D-RMSE) under imperfect depth map supervision. Ablation studies show the necessatiy of explicit geometric supervision and affine alignment when handling imperfect depth prior.

**Weaknesses:**

- The method description in Section 3.3 may be hard to follow for readers not familiar with neural rendering or SLAM, considering the medical imaging community MIDL serves. I would suggest the following clarifications:

	- Explain why using the disparity space and why inversion happens afterward.
	- Explain the concept of TSDF and its role in initialization.
	- Explain why normal consistency loss is necessary.

- Limited evaluation on in vivo data. The evaluation was done on phantom colonoscopy datasets, which provide well-controlled conditions and do not capture the complexity of in vivo colonoscopy (e.g., fluids, debris, strong non-rigid motion, etc.).

- All experiments assume ground-truth camera poses are known, and pose estimation is not jointly optimized. All authors may evaluate how sensitive GutSee is to realistic pose noise.

- Sensitivity to depth prior quality is not fully explored. Although the affine-invariant loss mitigates global bias, the method still relies on reasonable relative depth structure from the monocular prior. The paper does not analyze failure modes when the prior contains strong local errors.

- Runtime, memory usage, and scalability to longer sequences are not reported.

- There is no quantitative ablation isolating the contribution of the explicit BRDF and coaxial spotlight, beyond qualitative examples.

**Detailed Comments:**

See Weaknesses.

**Justification Of Final Rating:**

The revision improves the paper's quality in terms of writing quality and ablation studies. However, since the method is evaluated in a well-controlled environment with known camera poses, I believe its practicality in real-world settings remains debatable. Therefore, I will keep my original rating.

**Justification Of The Preliminary Rating:**

This paper presents a technically sound approach to a challenging problem in monocular medical image reconstruction.

The proposed affine-invariant depth supervision and explicit modeling of geometry and illumination address limitations of prior neural rendering methods and demonstrate convincing improvements under biased depth supervision from an off-the-shelf model, which is a realistic setting for endoscopy.

While the method description could be clearer for non-experts and the experimental validation is limited to phantom data with known camera poses, these issues primarily affect accessibility and generalization rather than the core scientific contribution.

Overall, the paper offers a meaningful methodological advance with clear potential value to the MIDL community, justifying a weak accept.

**Questions To Address In The Rebuttal:**

- Additional ablation studies as listed in Weaknesses would strengthen the paper.
- Better clarity in Section 3.3 for non-experts, as mentioned in Weaknesses.

---

> ### Author Response · Authors · 2026-01-24
> **Methodological Clarity and Additional Ablations**
>
> We thank Reviewer pe3H for their review and address each of their points below.
>
> **Q: Explain the use of disparity space and why inversion happens afterward.**
>
> We use disparity rather than depth because monocular depth estimators typically produce predictions that are affine-ambiguous in disparity space. Aligning rendered and predicted disparities via a per-frame affine transform yields a well-conditioned least-squares problem. After alignment, we invert the disparity to recover depth for geometric supervision. We have revised Section 3.3 to explicitly motivate this choice and clarify the inversion step for readers less familiar with neural rendering and depth estimation.
>
> **Q: Explain the concept of TSDF and its role in initialization.**
>
> In our pipeline, Truncated Signed Distance Function (TSDF) fusion produces a coarse initial surface estimate from monocular depth priors, providing a stable starting point for triangle optimization. It does not impose hard geometric constraints during optimization. All evaluated baselines use this same initialization, with the exception of EndoGSLAM and Endo-2DTAM, which are online methods initialized from the first depth frame in each sequence. We have added a brief explanation of TSDF fusion and its role as an initialization strategy in Section 3.3 to improve accessibility for a medical imaging audience.
>
> **Q: Explain why the normal consistency loss is necessary.**
>
> The normal consistency loss encourages local surface orientation to remain consistent with the depth prior after affine alignment, helping stabilize geometry in regions where photometric cues are weak or ambiguous. We have clarified the motivation for this term in Section 3.3 and described its role in complementing depth supervision.
>
> **Q: Limited evaluation on in-vivo data.**
>
> We focus on phantom datasets because they provide reliable ground-truth geometry essential for quantitative evaluation. Such ground truth is unavailable for in-vivo colonoscopy, making rigorous validation substantially more difficult. We view phantom-based evaluation as a necessary first step for geometry-centric reconstruction methods and have clarified this scope in Section 5.
>
> **Q: Assumption of known camera poses.**
>
> We assume known camera poses to isolate reconstruction quality from pose estimation errors and to focus specifically on the effects of biased monocular depth priors and explicit illumination modeling. Evaluating robustness to pose noise and extending GutSee to joint pose-geometry optimization are important directions for future work, which we have acknowledged in Section 5.
>
> **Q: Sensitivity to locally incorrect depth priors.**
>
> We have added an ablation study in Appendix E that evaluates this sensitivity. By introducing controlled local depth perturbations (Gaussian noise to randomly selected patches in a fraction p of frames), we found that GutSee is relatively insensitive to these errors: even with 90% of depth maps perturbed, photometric quality remains consistent and depth RMSE stays below 5 mm.
>
> **Q: Runtime, memory usage, scalability.**
>
> Our primary focus is on geometric reconstruction accuracy rather than computational efficiency, and we do not claim real-time or online feasibility. We therefore did not include detailed runtime or memory benchmarks. That said, our per-scene training time is comparable to other neural rendering methods.
>
> **Q: Quantitative ablation of BRDF and coaxial spotlight.**
>
> We have added a comprehensive ablation study in Appendix D examining both the complete illumination model and its individual components. Removing the BRDF model entirely increased depth RMSE by 0.249 mm and Chamfer distance by 0.122 mm. Removing the specular term alone increased depth RMSE by 0.320 mm. Removing the spotlight model yielded slightly better geometric metrics (depth RMSE of 4.696 mm vs. 4.728 mm) but significantly degraded rendering quality (LPIPS increased from 0.247 to 0.324). Detailed results are in Appendix D.
>
> **Q: Additional ablations would strengthen the paper.**
>
> In response to this and similar feedback from other reviewers, we have added ablation studies addressing illumination modeling (Appendix D) and depth prior sensitivity (Appendix E), strengthening the empirical grounding of the paper.
>
> ---
>
> In response to the reviewer's feedback, we have added ablation studies on illumination modeling (Appendix D) and depth prior sensitivity (Appendix E), and revised Section 3.3 to clarify our use of disparity space, TSDF initialization, and normal consistency loss. Given these additions, we kindly ask the reviewer to consider reassessing their score.

---

### Official Review · Reviewer_geM5 · 2026-01-10

**Confidence:** 4
**Preliminary Rating:** 4
**Final Rating:** 4

**Summary:**

The paper introduces GutSee, a depth-guided triangle splatting framework for monocular colonoscopic reconstruction that addresses two key challenges:
1. Affine ambiguity in monocular depth priors.
2.  Strong view-dependent specularities caused by coaxial endoscopic illumination.

- The method combines an **affine-invariant geometric supervision scheme** with a **physically motivated illumination model** incorporating an explicit coaxial spotlight and learnable BRDF parameters.
- By decoupling lighting effects from geometry and enforcing surface continuity through triangle primitives, GutSee achieves reconstructions that are both geometrically accurate and photometrically consistent.

Experiments on phantom colonoscopy datasets (C3VD v1 and v2) show that GutSee significantly improves geometric accuracy under biased depth supervision while maintaining competitive rendering quality, demonstrating robustness to imperfect monocular depth priors.

**Strengths:**

1. The paper clearly articulates why monocular colonoscopic reconstruction is uniquely challenging and why geometric fidelity.
2. The paper also has a strong experimental design; the comparison against multiple baselines under both ground-truth and biased supervision settings is fair and informative, and the inclusion of C3VDv2 increases the relevance of the evaluation.
3. *Appropriate use of triangle splatting*: Leveraging triangle primitives provides explicit surface normals and continuity, which naturally aligns with the lighting model and geometric supervision.
4. The affine-invariant depth supervision is a principled and elegant solution to per-frame scale and shift ambiguity in monocular depth estimators, *avoiding the pitfalls of enforcing incorrect metric constraints*.
5. Here explicitly modeling the coaxial endoscope light and using a BRDF-based reflectance model is well justified and effectively prevents specular highlights from being misinterpreted as geometry.

**Weaknesses:**

1. **Limited Ablation Scope**: Although an ablation study is provided, it is restricted to a single sequence; broader ablations across multiple scenes would strengthen confidence in the generality of the findings.

2. **Dependence on known camera poses**: The method assumes accurate camera poses, which isolates mapping quality but leaves open questions about performance in a full SLAM setting over long trajectories.

3. **Evaluation limited to phantom data**: While phantom datasets are appropriate for controlled evaluation, the absence of in-vivo experiments limits conclusions about robustness under real clinical conditions with fluids, debris, and strong non-rigid motion.

**Detailed Comments:**

The following suggestions can improve the paper slightly

1. Can you include a brief discussion comparing triangle splatting to recent geometry-regularized Gaussian splatting variants?
2. Would you consider reporting runtime or memory usage relative to baselines? This can improve practical realizations?
3. Can it also be clarified how sensitive the affine alignment is to severe local depth errors?

**Justification Of Final Rating:**

Thanks to the authors for a thorough and constructive rebuttal and for various clarifications and additions to the manuscript. Overall several of the concerns have been convincingly addressed.

* In particular, the newly added ablation studies on illumination modeling (Appendix D) and sensitivity to locally incorrect depth priors (Appendix E) directly respond to questions raised in the original review and strengthen the technical case for the proposed design.
    * The depth perturbation analysis provides useful empirical evidence that the affine-invariant supervision is robust to localized errors in monocular depth priors, which was previously unclear.
    * The expanded baseline comparisons and corrected evaluation pipeline further improves the fairness and credibility of the experimental results.

* The discussion of phantom-only evaluation, pose assumptions, and lack of downstream task validation has also been clarified appropriately.
   *  While these limitations remain, they are now clearly framed as scope choices rather than oversights, and the justification for focusing on phantom data with reliable ground-truth geometry is reasonable at this stage of the work.

**Overall, the rebuttal satisfactorily addresses the primary technical questions raised in the original review.**

*The remaining limitations—absence of in-vivo validation, reliance on known camera poses, and lack of task-level clinical evaluation are acknowledged and largely inherent to the current problem formulation rather than deficiencies that could be resolved within this submission.*

1. My overall assessment remains unchanged.
2. The paper represents a technically sound & well-motivated contribution that advances geometry-centric colonoscopic reconstruction under biased monocular depth supervision, and it remains appropriate for acceptance.

**Justification Of The Preliminary Rating:**

1. Overall this paper presents a **technically solid and well-motivated contribution** that advances geometry-centric reconstruction for colonoscopy.
2. The affine-invariant depth supervision and explicit illumination modeling address real and well-known failure modes in existing neural rendering approaches, particularly under biased monocular depth priors.
3. The experimental results convincingly demonstrate improved geometric accuracy without sacrificing rendering quality, and the discussion is honest about limitations.

Although, the lack of in-vivo evaluation and full SLAM integration prevents a strong accept, the paper is clearly above the acceptance threshold and represents valuable progress for the MIDL community.

**Questions To Address In The Rebuttal:**

1. How do the authors expect GutSee to behave under in-vivo conditions with significant non-rigid deformation and fluids?
1.a Are there plans for preliminary clinical validation?
2. How sensitive is the affine-invariant supervision to locally incorrect depth priors, and could failure modes be characterized more explicitly?
3. Do the authors anticipate any challenges in extending GutSee to joint pose and geometry optimization in a SLAM setting?
4. Can it also be empirically assess how improved geometry translates to better navigation, lesion localization, or measurement accuracy?

---

> ### Author Response · Authors · 2026-01-24
> **In-Vivo Generalization and Depth Prior Robustness**
>
> We thank Reviewer geM5 for their insightful comments. We address each point below.
>
> **Q: How do the authors expect GutSee to behave under in-vivo conditions with significant non-rigid deformation and fluids? Are there plans for preliminary clinical validation?**
>
> Achieving geometrically accurate 3D reconstruction on phantom data is a necessary first step before tackling in-vivo settings. The C3VD dataset shares key characteristics with real colonoscopy, including realistic colon geometry, endoscopic lighting conditions, and camera motion patterns, allowing us to meaningfully differentiate methods and identify which approaches are currently best suited for eventual clinical deployment. Notably, phantom datasets provide reliable ground-truth geometry essential for rigorous quantitative evaluation, which is unavailable for in-vivo data. Future work will expand to the in-vivo domain after demonstrating robust performance on phantom-based benchmarks. We have clarified this scope in Section 5.
>
> **Q: How sensitive is the affine-invariant supervision to locally incorrect depth priors, and could failure modes be characterized more explicitly?**
>
> To address this, we have added an ablation study in Appendix E that explicitly evaluates the effect of localized depth prior errors on reconstruction performance. We introduced controlled local perturbations to the monocular depth priors by adding Gaussian noise to randomly selected patches in a fraction p of frames. The results show that GutSee is relatively insensitive to these local errors: even when 90% of depth maps are perturbed, photometric quality remains consistent and depth RMSE stays below 5 mm. This robustness suggests that, given a sufficiently good initialization, local errors in the depth supervision do not substantially degrade reconstruction quality.
>
> **Q: Do the authors anticipate any challenges in extending GutSee to joint pose and geometry optimization in a SLAM setting?**
>
> Extending GutSee to joint pose and geometry optimization in a SLAM setting is an important direction for future work. In this study, we assume known camera poses to focus specifically on the effects of biased monocular depth priors and explicit illumination modeling on reconstruction quality. This allows us to disentangle reconstruction performance from pose estimation errors and isolate the contributions of our geometric and photometric modeling components. Incorporating joint pose optimization would introduce additional challenges, including sensitivity to pose drift under non-Lambertian lighting and over long trajectories. We have acknowledged these challenges and outlined SLAM integration as future work in Section 5.
>
> **Q: Can it also be empirically assessed how improved geometry translates to better navigation, lesion localization, or measurement accuracy?**
>
> To the best of our knowledge, C3VD does not provide the annotations necessary for direct evaluation of navigation, lesion localization, or measurement accuracy. Our goal in this work is to first establish geometrically faithful monocular colonoscopic reconstruction, which is a necessary prerequisite before downstream clinical applications can be reliably evaluated.
>
> ---
>
> In response to the reviewer's feedback, we have added an ablation study on depth prior sensitivity (Appendix E) and expanded our discussion of future directions including in-vivo evaluation and SLAM integration (Section 5). Given these revisions, we kindly ask the reviewer to consider increasing their score.

---

> > ### Comment · Reviewer_geM5 · 2026-01-24
> > **Response to Rebuttal**
> >
> > Thanks to the authors for a thorough and constructive rebuttal and for various clarifications and additions to the manuscript. Overall several of the concerns have been convincingly addressed.
> >
> > * In particular, the newly added ablation studies on illumination modeling (Appendix D) and sensitivity to locally incorrect depth priors (Appendix E) directly respond to questions raised in the original review and strengthen the technical case for the proposed design.
> >     * The depth perturbation analysis provides useful empirical evidence that the affine-invariant supervision is robust to localized errors in monocular depth priors, which was previously unclear.
> >     * The expanded baseline comparisons and corrected evaluation pipeline further improves the fairness and credibility of the experimental results.
> >
> > * The discussion of phantom-only evaluation, pose assumptions, and lack of downstream task validation has also been clarified appropriately.
> >    *  While these limitations remain, they are now clearly framed as scope choices rather than oversights, and the justification for focusing on phantom data with reliable ground-truth geometry is reasonable at this stage of the work.
> >
> > **Overall, the rebuttal satisfactorily addresses the primary technical questions raised in the original review.**
> >
> > *The remaining limitations—absence of in-vivo validation, reliance on known camera poses, and lack of task-level clinical evaluation are acknowledged and largely inherent to the current problem formulation rather than deficiencies that could be resolved within this submission.*
> >
> > 1. My overall assessment remains unchanged.
> > 2. The paper represents a technically sound & well-motivated contribution that advances geometry-centric colonoscopic reconstruction under biased monocular depth supervision, and it remains appropriate for acceptance.

---

### Official Review · Reviewer_phut · 2026-01-15

**Confidence:** 4
**Preliminary Rating:** 4
**Final Rating:** 4

**Summary:**

This paper proposes GutSee, a triangle-splatting–based reconstruction framework for monocular colonoscopy that combines affine-invariant depth supervision with an explicit, physically motivated endoscopic illumination model. By aligning rendered depth to monocular priors up to per-frame scale and shift, the method leverages imperfect depth estimates without inheriting their bias, while the coaxial spotlight and learnable BRDF prevent specular highlights from corrupting geometry. Experiments on phantom colonoscopy datasets show that the made method achieves substantially improved geometric accuracy under biased depth supervision compared to prior methods, while maintaining competitive rendering quality. The results highlight the importance of explicitly modeling both depth ambiguity and endoscopic lighting to obtain geometrically reliable reconstructions for downstream clinical use.

**Strengths:**

1) This paper targets a realistic setup where scale-known and accurate depth priors are unavailable. Tackling this limitation can further enhance the clinical applicability of endoscopic scene reconstruction methods.
2) Illumination inconsistency is a unique and severe problem in endoscopic scene reconstruction, and the authors put efforts into improving the relevant techniques.
3) The geometric improvement of the made method is promising.
4) Introducing an illumination model to assist geometric refinement is a constructive direction for advancing endoscopic scene reconstruction

**Weaknesses:**

1) The authors claimed that the proposed “physically motivated lighting model” can “prevent specular reflections from introducing spurious geometric detail.” However, the ablation of this illumination model is lacking, particularly in terms of its effectiveness in preserving geometric fidelity.
2) The authors introduce a new scene representation (triangle splatting), which differs from existing works using 2D/3D or surfel-like GS, lacking a well-motivated elaboration or experiment on the superiority of the chosen representation.
3) SLAM-based method is incorporated as a comparison method. Endo-2DTAM, a SLAM method indicating much better geometric accuracy, is overlooked in both the experiment and the reference.

**Detailed Comments:**

I suggest the authors to add the experiment details or clarifications per weakness given above.

**Justification Of Final Rating:**

This work demonstrates the great effectiveness of improving geometric performance under scale-unknown depth priors, indicating meaningful and potential use in a clinical setup. The authors solve all my concerns during rebuttal, I decide to keep my score

**Justification Of The Preliminary Rating:**

Despite insufficient comparison and elaboration on the developed method, this work demonstrates the great effectiveness of improving geometric performance under scale-unknown depth priors, indicating meaningful and potential use in a clinical setup.

**Questions To Address In The Rebuttal:**

1) Can the authors provide quantitative or ablation-based evidence demonstrating how the illumination model contributes to preserving geometric fidelity? In particular, how does geometry quality change when the illumination model (or specific components such as the spotlight attenuation or BRDF terms) is removed or simplified?
2) How does the proposed model compare to simpler lighting modeling strategies in terms of geometry accuracy?
3) What are the concrete advantages of triangle splatting for endoscopic reconstruction in this setting? Can the authors provide conceptual justification or empirical comparison showing why triangle splatting is more suitable (e.g., in terms of geometric stability, normal estimation, or interaction with illumination modeling) than GS-based or surfel-based representations?
4) Can the authors clarify why Endo-2DTAM was excluded from the evaluation? If inclusion is not feasible, can the authors provide a discussion or indirect comparison to contextualize how their method relates to or improves upon Endo-2DTAM in terms of geometric accuracy and assumptions?

---

> ### Author Response · Authors · 2026-01-24
> **Illumination Modeling and Baseline Comparisons**
>
> We thank Reviewer phut for their detailed review. We address each point below.
>
> **Q: Can the authors provide quantitative or ablation-based evidence demonstrating how the illumination model contributes to preserving geometric fidelity? In particular, how does geometry quality change when the illumination model (or specific components such as the spotlight attenuation or BRDF terms) is removed or simplified?**
>
> We have added a comprehensive ablation study in Appendix D examining both the complete illumination model and its individual components. Removing the BRDF model entirely increased depth RMSE by 0.249 mm and Chamfer distance by 0.122 mm across our four test scenes, while also degrading photometric accuracy (LPIPS increased from 0.247 to 0.269). To further isolate individual contributions, we ablated the specular BRDF term and spotlight attenuation separately. Removing the specular term increased depth RMSE by 0.320 mm, supporting our claim that accurate illumination modeling improves geometric fidelity. Removing the spotlight model yielded slightly better geometric metrics than our base configuration (depth RMSE of 4.696 mm vs. 4.728 mm); however, this comes at a significant cost to rendering quality, with LPIPS increasing from 0.247 to 0.324. This degradation occurs because the spotlight model is necessary to accurately capture distance-dependent lighting effects.
>
> **Q: How does the proposed model compare to simpler lighting modeling strategies in terms of geometry accuracy?**
>
> We evaluated GutSee under several simplified lighting configurations, including diffuse-only shading and removing the explicit coaxial spotlight model. Across all test sequences, these simplified variants either produced higher depth RMSE and Chamfer distance compared to our full model, or significantly degraded rendering quality. Detailed results are provided in Appendix D.
>
> **Q: What are the concrete advantages of triangle splatting for endoscopic reconstruction in this setting? Can the authors provide conceptual justification or empirical comparison showing why triangle splatting is more suitable (e.g., in terms of geometric stability, normal estimation, or interaction with illumination modeling) than GS-based or surfel-based representations?**
>
> Our choice of triangle splatting is motivated by both conceptual and practical considerations. First, triangles are a native primitive in mesh-based rendering and surface reconstruction pipelines, keeping GutSee closely aligned with standard surface representations and facilitating integration with mesh-based analysis, visualization, and downstream clinical tools. Second, the surface normals of triangles can be computed analytically from vertex positions, which is advantageous for our physically based illumination modeling; in contrast, Gaussian- or surfel-based representations require implicit or approximate normal estimation that can be less stable. Third, unlike Gaussian splatting methods whose kernels have infinite spatial support, triangle splats have bounded support, reducing ambiguity about where the reconstructed surface lies. Finally, the use of triangle primitives naturally positions GutSee for future extensions toward explicit mesh reconstruction of colon scenes, an important direction for enabling clinically meaningful surface analysis beyond novel view synthesis. We have added language motivating our use of triangle splatting in Section 3.1.
>
> **Q: Can the authors clarify why Endo-2DTAM was excluded from the evaluation? If inclusion is not feasible, can the authors provide a discussion or indirect comparison to contextualize how their method relates to or improves upon Endo-2DTAM in terms of geometric accuracy and assumptions?**
>
> In the original submission, Endo-2DTAM was excluded due to implementation constraints at the time of evaluation. We agree that its inclusion strengthens the empirical comparison. In response to this feedback, we have added both Endo-2DTAM and 2DGS to our set of baselines, reporting their performance alongside existing methods under the same evaluation protocol (Table 1). The inclusion of these additional baselines does not change our main results: GutSee continues to achieve the best geometric reconstruction accuracy across all evaluated sequences under biased supervision.
>
> ---
>
> In response to the reviewer's feedback, we have added ablation studies examining our illumination model (Appendix D) and included Endo-2DTAM and 2DGS as baselines (Table 1). Given these revisions, we respectfully ask the reviewer to consider increasing their score.

---

> > ### Comment · Reviewer_phut · 2026-01-26
> > **Response to rebuttal**
> >
> > Overall, the rebuttal satisfactorily addresses my concerns. I will keep my score

---

### Author Rebuttal · Authors · 2026-01-24

**Rebuttal:**

We thank all reviewers for their constructive feedback. In response, we have made the following revisions to strengthen the manuscript:

**Expanded Baselines.** We added 2DGS and Endo-2DTAM to our evaluation (Table 1). GutSee continues to achieve the best geometric reconstruction accuracy across all sequences under biased supervision.

**Corrected Evaluation.** We identified and corrected inconsistent preprocessing parameters across sequences, ensuring all sequences now use the same standardized pipeline. We also increased the depth loss weight for 3DGS to ensure fair comparison. All methods were re-evaluated on the corrected data; our main conclusions remain unchanged.

**New Ablation Studies.** We added two comprehensive ablation studies:
- Appendix D examines our illumination model, demonstrating that accurate lighting modeling improves geometric fidelity.
- Appendix E tests sensitivity to locally incorrect depth priors, demonstrating that GutSee remains robust to localized perturbations in depth supervision.

**Improved Clarity.** We revised Section 3.3 to motivate our use of disparity space, TSDF initialization, and normal consistency loss. We also added discussion in Section 3.1 justifying our choice of triangle splatting over Gaussian or surfel representations.

**Scope and Future Work.** We clarified in Section 5 that phantom-based evaluation provides essential ground-truth geometry unavailable in vivo, and acknowledged SLAM integration and in-vivo extension as important future directions.

We believe these revisions address the reviewers' primary concerns and respectfully ask that they consider reassessing their scores.

**Supporting Material:**

/attachment/8077cfd94c30afb55d4b615a4377a6cbbdaa5ec7.zip

---

### Meta-Review · Area_Chair_Tt85 · 2026-02-07

**Recommendation:** Accept (Poster)
**Confidence:** 5

**Metareview:**

The manuscript received detailed and overall positive initial assessments that persist through revisions.
During revisions, the authors addressed some of the technical concerns through ablation studies and robustness analyses, particularly regarding illumination modeling and depth prior sensitivity. While the current evaluation remains focused on well-controlled phantom environments with known camera poses -- which continues to be perceived as a considerable limitation -- the work is considered a technically sound and well-motivated framework that meaningfully advances geometry-centric reconstruction under biased monocular supervision. Overall assessment is that this work can be accepted.

---

### Decision · Program_Chairs · 2026-02-13

Accept (Poster)